# OpenGaussian: Towards Point-Level 3D Gaussian-based Open Vocabulary Understanding

**Yanmin Wu**[1,2]   **Jiarui Meng**[1]   **Haijie Li**[1]   **Chenming Wu**[3]*  **Yahao Shi**[4]   **Xinhua Cheng**[1]

**Chen Zhao**[3]   **Haocheng Feng**[3]   **Errui Ding**[3]   **Jingdong Wang**[3]   **Jian Zhang**[1,2]*

[1] School of Electronic and Computer Engineering, Peking University
[2] Guangdong Provincial Key Laboratory of Ultra High Definition Immersive Media Technology,
Shenzhen Graduate School, Peking University
[3]Baidu VIS        [4]Beihang University

## Abstract

This paper introduces OpenGaussian, a method based on 3D Gaussian Splatting (3DGS) capable of 3D point-level open vocabulary understanding. Our primary motivation stems from observing that existing 3DGS-based open vocabulary methods mainly focus on 2D pixel-level parsing. These methods struggle with 3D point-level tasks due to weak feature expressiveness and inaccurate 2D-3D feature associations. To ensure robust feature presentation and 3D point-level understanding, we first employ SAM masks without cross-frame associations to train instance features with 3D consistency. These features exhibit both intra-object consistency and inter-object distinction. Then, we propose a two-stage codebook to discretize these features from coarse to fine levels. At the coarse level, we consider the positional information of 3D points to achieve location-based clustering, which is then refined at the fine level. Finally, we introduce an instance-level 3D-2D feature association method that links 3D points to 2D masks, which are further associated with 2D CLIP features. Extensive experiments, including open vocabulary-based 3D object selection, 3D point cloud understanding, click-based 3D object selection, and ablation studies, demonstrate the effectiveness of our proposed method. The source code is available at our project page https://3d-aigc.github.io/OpenGaussian.

## 1   Introduction

The recently proposed neural rendering method 3D Gaussian Splatting (3DGS) [19] has rapidly gained popularity and is being widely applied in various areas such as 3D reconstruction [18, 10, 38], 4D reconstruction [23, 28, 42, 47], generation [41, 25, 49], and understanding [44, 51]. This is primarily due to its fast training, real-time rendering capabilities, and explicit point-based representation. Within the realm of 3D vision learning, 3D scene understanding has embraced the 3DGS framework to achieve an integrated process encompassing explicit reconstruction, novel view synthesis, and semantic understanding. Incorporating open vocabulary for 3D understanding is considered a more promising, practical, and natural approach for intelligent agent understanding, interaction, and decision-making. In this paper, we specifically concentrate on point-level open-vocabulary 3D scene understanding based on the 3DGS framework.

Despite several efforts [33, 37, 52, 48, 54, 15, 24] to incorporate learnable language attributes into 3DGS, aiming to enhance them with language-grounded capabilities, the primary objective of these

---

*Corresponding authors.

38th Conference on Neural Information Processing Systems (NeurIPS 2024).

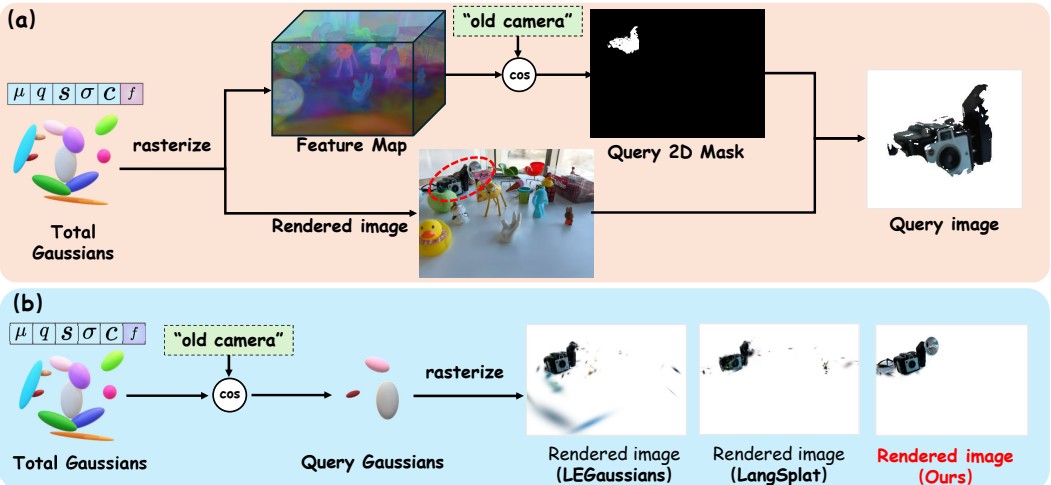

Figure 1: Illustration of two text query strategies. (a) demonstrates the process of rendering a feature map and computing its similarity with text features to obtain a 2D mask, which is then used to generate a corresponding rendered image. (b) demonstrates the direct similarity computation of 3D Gaussian language features with text features, selecting Gaussian points with high similarity, and rendering to obtain a rendered image corresponding to the text.

approaches is to render language attributes onto images for 2D pixel-level understanding, lifting 2d feature into view-consistent understanding/segmentation. As shown in Fig. 1(a), we demonstrate this using an example involving the rendering of language-embedded 3D Gaussians into 2D feature maps using LangSplat [33], followed by the utilization of open vocabulary for matching and identifying target areas. While these methods exhibit impressive performance in view-consistent lifting, we find a few drawbacks such as: **1)** inability to recognize occluded objects or parts, compromising the inherent 3D capabilities of 3DGS; **2)** incompatibility with robotics and embodied intelligence applications that necessitate 3D point-level understanding, localization, and interaction. Therefore, we aim to empower 3DGS with the capability of **3D point-level open-vocabulary understanding**.

We first investigate the 3D point-level understanding of existing works. As depicted in Fig 1(b), we measure the similarity between textual features and 3D Gaussian language features, selecting points with high relevance, and subsequently render these points through the rasterization module of 3DGS to generate images. The results highlight challenges in effectively matching target objects, indicating that although these approaches exhibit strong performance on 2D images, their 3D understanding capabilities are limited. As demonstrated in Fig. 6, our visualization of their 3D point features reveals a lack of discriminability between different objects and low consistency within objects. We attribute these limitations to two main factors: **1) Weak feature expressiveness**: Due to the memory and speed constraints associated with 3D point-wise training and rendering in 3DGS, training high-dimensional language features for millions of Gaussian points in a scene becomes challenging. As a result, existing methods rely on dimension reduction techniques such as distillation [52, 8] or quantization [37] to reduce dimensions. However, this inevitably compromises the expressiveness and distinguishability of the features. **2) Inaccurate 2D-3D correspondence**: The alpha-blending rendering technique accumulates the values of 3D points based on opacity weights to render 2D pixels, which prevents the establishment of a one-to-one correspondence between 2D and 3D. Consequently, a performance mismatch occurs between 2D and 3D interpretations.

To address these challenges, we propose OpenGaussian, an approach that learns distinctive and consistent features at the 3D point-level, both across objects and within objects. Our method associates high-dimensional lossless CLIP [34] features with 3D Gaussian points, enabling open-vocabulary 3D scene understanding. Specifically, the technical contributions of this paper are summarized as: **1)** Training of 3D point-level instance features that are both distinctive and 3D consistent using the proposed intra-mask smoothing loss and inter-mask contrastive loss, leveraging boolean masks from SAM [21] without cross-frame associations; **2)** Introducing a two-level coarse-to-fine codebook to discretize the instance features, resulting in discrete 3D instance clusters. **3)** Proposing an instance-level 2D-3D association method based on IoU and feature distance to associate CLIP features from

multiple views for each 3D instance. Through comprehensive experiments covering open-vocabulary object selection at the 3D point level, open-vocabulary 3D point cloud understanding, click-based 3D object selection, and module ablation, we demonstrate the simplicity and efficiency of our method. OpenGaussian eliminates the need for an additional network for feature dimensionality compression or quantization while inheriting the open-vocabulary capabilities of the original CLIP features.

## 2    Related Work

### 2.1    Neural Rendering

Neural 3D scene representation, for example, the recently proposed NeRF [29] has demonstrated remarkable advancements in novel view synthesis quality using learning-based optimization techniques. While many methods focus on improving NeRF's rendering quality [2, 4, 3, 17], they often suffer from slow training and rendering speeds. Alternatively, methods based on explicit representation, such as voxels [39, 14, 35], hash grids [30] and point clouds [50, 1, 36], have emerged. These methods employ techniques to reduce the computational cost of large neural networks. The recent development of 3DGS [19] sets a new benchmark in terms of both rendering quality and rendering speed by employing fast differentiable rasterization of 3D Gaussians instead of volume rendering. As neural rendering proves to be an effective connection between 2D images and 3D scenes, our work builds upon this paradigm with a particular focus on 3D point-level open-vocabulary understanding.

### 2.2    3D Open Vocabulary Understanding

Recent advancements in open vocabulary scene understanding have seen the integration of 2D Vision-Language Models (VLMs) with 3D point cloud processing, resulting in significant progress in the field [16, 45, 53]. These approaches primarily concentrate on aligning features and projecting 3D data into 2D, thereby enhancing zero-shot learning capabilities. Furthermore, significant progress has been made in 3D object detection and segmentation [11, 27, 40], demonstrating the efficacy of merging point cloud data with visual features extracted from images for scene analysis.

The significant advancements in 2D scene understanding, pioneered by SAM [21] and its variants, have motivated the exploration of integrating semantic features into NeRF. Methods have been developed to incorporate semantic features from models such as CLIP [34] and DINO [7] into NeRF, enabling more effective handling of 3D segmentation, understanding, and editing tasks. LERF [20] distills features from readily available VLMs like CLIP into a 3D scene represented by NeRF. [26] also introduces a 3D open-vocabulary segmentation pipeline using NeRF. Recent efforts have been made to combine 2D scene understanding techniques with 3D Gaussians to create a real-time and editable 3D scene representation, addressing the computational challenges of NeRF-based methods.

LEGaussians [37] introduces uncertainty and semantic feature attributes to each Gaussian, to render a semantic map with corresponding uncertainties. This rendered map is compared with the quantized CLIP and DINO dense features extracted from the ground truth image. LangSplat [33] uses a scene-wise language autoencoder to learn language features on the scene-specific latent space, demonstrating to discern clear boundaries between objects in rendered feature images. Feature3DGS [52] proposes a parallel $N$-dimensional Gaussian rasterizer to distill high-dimensional features for view-based tasks such as editing and segmentation. To achieve the 2D mask consistency across views, Gaussian Grouping [48] performs simultaneous reconstruction and segmentation of open-world 3D objects, guided by 2D mask predictions obtained from SAM and 3D spatial consistency constraints. Similar to these works, we leverage the real-time rendering and explicit representation capabilities of 3DGS. However, while those methods primarily focus on pixel-level open-vocabulary understanding (*i.e.*, lifting 2D feature into view-consistent segmentation), our approach diverges as we aim to enhance 3DGS with the ability for 3D point-level open-vocabulary understanding.

## 3    Method

### 3.1    3D Consistency-Preserving Instance Feature Learning

3DGS [19] utilizes an explicit scene representation through 3D Gaussian points. Each Gaussian point encompasses various attributes such as position $\boldsymbol{\mu}$, rotation $\boldsymbol{R}$, scale $\boldsymbol{S}$, opacity $\sigma$, and spherical

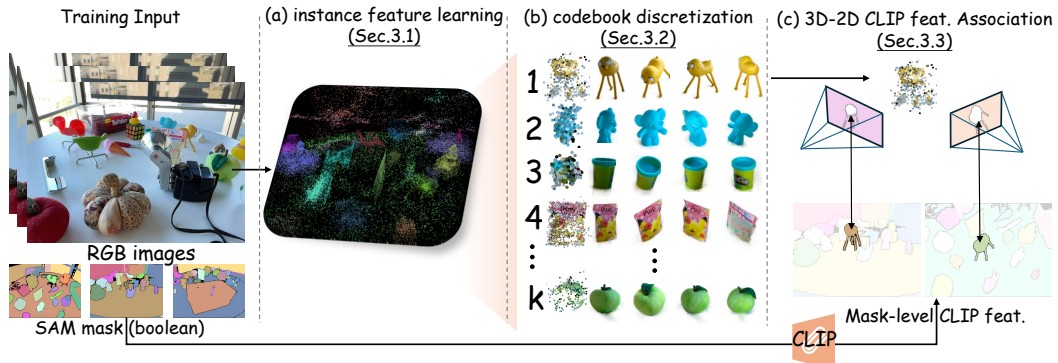

Figure 2: (a) We use the view-independent SAM boolean mask to train 3D instance features with 3D consistency for 3DGS. (b) We propose a two-level codebook for discretizing instance features from coarse to fine. (c) An instance-level 3D-2D feature association method to associate 2D CLIP features with 3D points without training.

harmonics coefficients for representing direction-aware color $c$. For a detailed description of the splatting process, please refer to Appendix A.3, where a set of 3D Gaussian points is projected onto 2D screen space and blended to generate pixels. Inspired from prior studies [33, 37, 52], we augment each 3D Gaussian point with a low-dimensional feature $f \in \mathbb{R}^6$ to represent its instance attributes. However, our approach differs in two crucial aspects: **1)** We do not require additional dimensional reduction, quantization, or distillation for pre-trained features (such as CLIP [34], SAM [21], DINO [7], and LSeg [22]) that widely used in previous literature [33, 52, 37, 8]; **2)** Instead of relying on tracking-based 2D methods for object counting in the scene [48], we exploit the multi-view global consistency of the 3D Gaussians to constrain the instance features. We adhere to the principle that Gaussian-rendered features from the same object should be close, while those from different objects should be distant. To achieve this, we employ binary SAM masks (instead of high-dimensional SAM features) without cross-view correlation to supervise the rendered instance feature maps using two types of losses: intra-mask smoothing loss and inter-mask contrastive loss.

Given an arbitrary training view, we follow the splatting process to render the 3D instance features $f$ into a feature map $M \in \mathbb{R}^{6 \times H \times W}$ by alpha-blending. Given the $i$-th SAM mask $B_i \in \{0, 1\}^{1 \times H \times W}$, we can obtain the mean feature within the mask: $\bar{M}_i = (B_i \cdot M) / \sum B_i \in \mathbb{R}^6$. To ensure that features within each mask are close to their mean, we introduce the **intra-mask smoothing loss**, which is defined as follows:

$$\mathcal{L}_s = \sum_{i=1}^{m} \sum_{h=1}^{H} \sum_{w=1}^{W} B_{i,h,w} \cdot \left\| M_{:,h,w} - \bar{M}_i \right\|^2, \tag{1}$$

where $H$ and $W$ represent the height and width of the image, respectively, while $m$ corresponds to the number of SAM masks in the current view. Additionally, we incorporate a constraint to promote feature diversity among different instances, increasing the mean feature distance between masks. This constraint is referred to as the **inter-mask contrastive loss**, and it can be described as follows:

$$\mathcal{L}_c = \frac{1}{m(m-1)} \sum_{i=1}^{m} \sum_{j=1,j \neq i}^{m} \frac{1}{\left\| \bar{M}_i - \bar{M}_j \right\|^2}, \tag{2}$$

where $m$ represents the number of masks, $\bar{M}_i$ and $\bar{M}_j$ denote the mean features of two distinct masks. By utilizing these strategies, we successfully obtain significant **3D cross-view consistency** and **distinct** instance features directly from masks, eliminating the need for cross-view correlation.

### 3.2 Two-Level Codebook for Discretization

Intuitively, the learned instance features appear well-suited for interactive 3D object segmentation. For instance, by clicking on a pixel within the rendered feature map, we can retrieve Gaussians with similar features to identify the selected object. However, practical implementation of this approach poses challenges for the following reasons: **1)** Setting a universal threshold to select similar

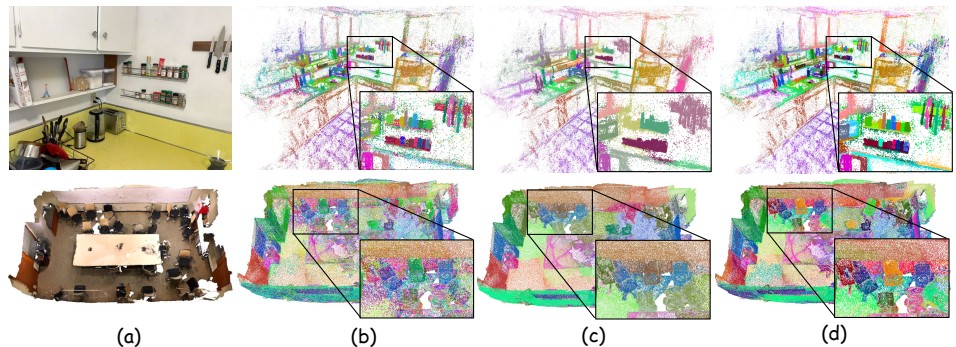

Figure 3: (a) Reference image/mesh; (b) instance features learned from Sec. 3.1; (c)-(d) Point features after discretization by coarse-level and fine-level codebook (Sec. 3.2).

features proves to be difficult; **2)** Since the feature map is rendered using alpha-blending, which accumulates weights, it is inevitable for Gaussians from the same object to exhibit dissimilar features, while Gaussians from different objects may share similar features. To enhance the distinctiveness of instance features and improve interactivity for downstream tasks, we aim to ensure that Gaussians from the same instance possess **identical (not just similar)** features by discretizing them. Inspired by prior works on 3DGS compression [31, 12], we propose employing codebook discretization to address this challenge. As depicted in Fig. 3(b), the point features before discretization exhibit noise.

**(1) Codebook for Discretization**. Given the instance features $F \in \mathbb{R}^{n \times 6}$ for all $n$ Gaussians, we first randomly select $k = 64$ features from $F$ to initialize the quantization codebook $C \in \mathbb{R}^{k \times 6}$. **1)** For each instance feature $\{f_i\}_{i=1}^n$, we find the closest quantized feature $\{c_j\}_{j=1}^k$ in the codebook $C$, and store each Gaussian's quantization index $j$ in $I \in \mathbb{R}^{n \times 1}$. **2)** In the forward process of feature map rendering and loss calculation, $c_j$ replaces $f_i$ in computations. **3)** During backpropagation, the gradients of the quantized features are copied to the instance features (*i.e.* $\frac{\partial \mathcal{L}_p}{\partial f_i} = \frac{\partial \mathcal{L}_p}{\partial c_j}$, $\mathcal{L}_p$ is defined in Eq. (4)), thus optimizing the instance features $f_i$. **4)** Subsequently, the quantization codebook $C$ is updated based on the indices $I$ and $F$. Steps 1) to 4) are then repeated. Finally, we transform the continuous instance features $F$ into quantized features and indices $\{C, I\}$, achieving discretization of instances in the scene.

However, this solution still presents challenges: **1)** Due to occlusions or distance, two objects may never share the same viewpoint and thus remain unoptimized by contrastive loss (*i.e.* Eq. (2)), failing to ensure their features are distinct. **2)** In large scenarios, a $k$ value of 64 proves inadequate for distinguishing all objects, reducing the distinctiveness of instance features. However, Simply increasing $k$ does not improve performance (will be demonstrated in experiments (Sec. 4.4)).

**(2) Two-Level Codebook**. We propose a two-level, coarse-to-fine codebook discretization to address the above issues. Initially, we concatenate the instance features $F$ with the 3D coordinates $X \in \mathbb{R}^{n \times 3}$ of the Gaussians for codebook construction, enabling position-dependent clustering. Subsequently, we further discretize within each coarse cluster based only on the instance features. Therefore, this approach not only avoids the issue of distantly located, non-co-visible objects being assigned to the same one, but also breaks down large scenes, reducing the complexity of optimization. The process can be mathematically expressed as:

$$\begin{cases} \left[F \in \mathbb{R}^{n \times 6}; X \in \mathbb{R}^{n \times 3}\right] \mapsto \{C_{\text{coarse}} \in \mathbb{R}^{k_1 \times (6+3)}, I_{\text{coarse}} \in \{1, \ldots, k_1\}^n\} & \text{coarse, } k_1 = 64, 32 \\ F \in \mathbb{R}^{n \times 6} \mapsto \{C_{\text{fine}} \in \mathbb{R}^{(k_1 \times k_2) \times 6}, I_{\text{fine}} \in \{1, \ldots, k_2\}^n\} & \text{fine, } k_2 = 10, 5 \end{cases}$$
(3)

Finally, we discretized the continuous instance features $F$ into a two-level codebook $\{C, I\}_{coarse}, \{C, I\}_{fine}$. Notably, at the coarse level, the position of the Gaussians is used solely for the codebook construction and is not involved in optimization, thus preserving the geometric structure of the pre-trained Gaussian model. The visualization results of the two-level codebook can be seen in Fig. 3(c) and (d).

**(3) Pseudo Feature Loss**. In the instance feature learning stage (Sec.3.1), supervision is limited to boolean masks. However, during the current codebook construction stage, we have obtained distinctive instance features that now serve as stronger supervision. Therefore, we can replace the previous mask losses (Eq. (1), (2)) and clone the instance features from the first stage as pseudo

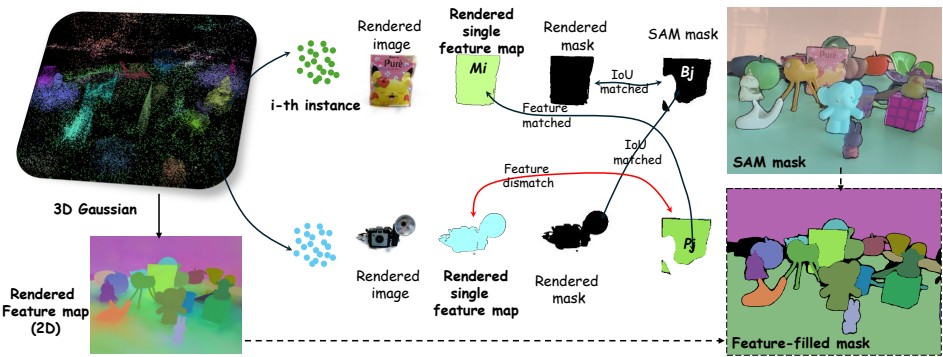

Figure 4: We render 3D instance points to an arbitrary training view, and associate 3D points with 2D masks based on the principle of joint IoU and feature similarity, which have already been extracted with mask-level CLIP features, thereby indirectly associating 3D points with CLIP features.

ground truth. The training objective becomes:

$$\mathcal{L}_p = \|\boldsymbol{M}_p - \boldsymbol{M}_c\|^1, \tag{4}$$

where $\boldsymbol{M_p} \in \mathbb{R}^{6 \times H \times W}$ is the feature map rendered from the first stage pseudo features, and $\boldsymbol{M_c} \in \mathbb{R}^{6 \times H \times W}$ represents the feature map rendered from quantized features.

### 3.3 Instance-Level 2D-3D Association without Depth Test

Through the codebook discretization process described above, we enhance the ability to select 3D objects via prompts such as clicking (will be demonstrated in Sec. 4.3). To further enable more natural, open-vocabulary-based interactions, it is essential to associate 3D Gaussians with language features effectively. In language-embedded 3D frameworks, there are two solutions: **1)** Compressing or distilling image features (already linked with linguistic features in the CLIP embedding space) into a lower dimension to train 3D Gaussian semantic fields, which requires additional networks, training steps, and potentially scene-specific encoder-decoders. Additionally, the compressed features may blur the original semantics. **2)** Establishing an association between 3D points and 2D pixels using camera intrinsic and extrinsic parameters to map image features onto 3D points, but necessitates depth information for occlusion testing [32].

We propose a simple yet efficient instance-level 3D-2D association method that retains high-dimensional, lossless linguistic features while avoiding the need for depth-based occlusion testing. Specifically, as shown in Fig. 4, **1)** we first render the features of a single 3D instance to the current view, called "single-instance map" $\boldsymbol{M}_i \in \mathbb{R}^{6 \times H \times W}$ (where $i$ is the 3D instance index, ranging from 1 to $k_1 \cdot k_2$), and then compute the Intersection over Union (IoU) with the current view's "SAM mask" $\boldsymbol{B}_j \in \{0, 1\}^{1 \times H \times W}$ (where $j$ is the mask index, ranging from 1 to the total masks.). Intuitively, the SAM mask with the highest IoU is associated with this 3D instance. However, due to occlusions, one "SAM mask" may intersect with a "single-instance map" rendered from multiple 3D instances, which is why the previously mentioned pixel-to-point association method requires depth for occlusion testing. **2)** Our solution populates boolean-type "SAM mask" $\boldsymbol{B}_j$ with pseudo GT features, termed "feature-filled mask" $\boldsymbol{P}_j \in \mathbb{R}^{6 \times H \times W}$, and then calculates the feature distance between $\boldsymbol{P}_j$ and $\boldsymbol{M}_j$, thus avoiding situations where IoU is high but the features do not correspond to the same object. In other words, we propose a unified criterion of IoU and feature distance, which can be formulated as:

$$\mathcal{S}_{ij} = \text{IoU}(\pi(\boldsymbol{M}_i), \boldsymbol{B}_j) \cdot (1 - \|\boldsymbol{M}_i - \boldsymbol{P}_j\|^1), \tag{5}$$

where $\mathcal{S}_{ij}$ represents the score between the $i$-th 3D instance and the $j$-th SAM mask in the current view. The first term calculates the IoU, with $\pi(\cdot)$ indicating the binarization operation; the second term's value is inversely proportional to the feature distance. Finally, the CLIP image features of the mask with the highest score are associated with the Gaussians of the 3D instance, and the integration of multi-view features is also considered.

Table 1: Performance of object selection in 3D space from text query on LeRF dataset. Accurate is measured by mAcc@0.25. `waldo_kitchen` abbreviated as `kitchen`.

| Methods | mIoU ↑ | | | | | mAcc. ↑ | | | | |
|---|---|---|---|---|---|---|---|---|---|---|
| | figurines | teatime | ramen | kitchen | **Mean** | figurines | teatime | ramen | kitchen | **Mean** |
| LangSplat [33] | 10.16 | 11.38 | 7.92 | 9.18 | 9.66 | 8.93 | 20.34 | 11.27 | 9.09 | 12.41 |
| LEGaussians [37] | 17.99 | 19.27 | 15.79 | 11.78 | 16.21 | 23.21 | 27.12 | 26.76 | 18.18 | 23.82 |
| **OpenGaussian** | **39.29** | **60.44** | **31.01** | **22.70** | **38.36** | **55.36** | **76.27** | **42.25** | **31.82** | **51.43** |

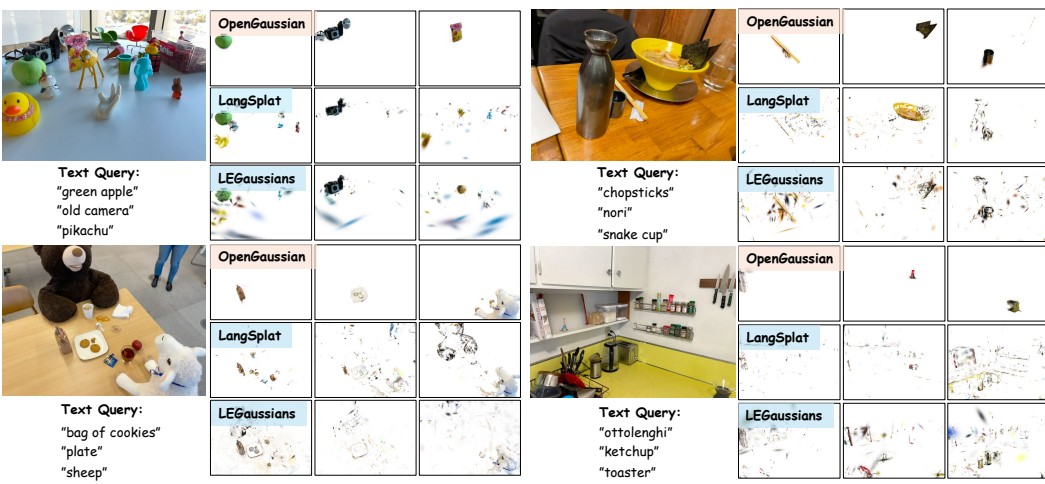

Figure 5: Open-vocabulary 3D object selection on the LERF dataset. OpenGaussian outperforms LangSplat and LEGaussians in accurately identifying the 3D objects corresponding to text queries.

## 4  Experiments

### 4.1  Open-Vocabulary Object Selection in 3D Space

**Settings. 1) Task**: Given an open-vocabulary text query, we extract its text feature using CLIP and calculate the cosine similarity between this feature and the language features of each Gaussian. Then, we select the highly relevant 3D points and render them into multi-view images using the 3DGS pipeline. **2) Baseline**: We compared our method with LangSplat and LEGaussians. OpenGaussian associates each Gaussian with a 512-dimensional CLIP feature using the method described in Sec. 3.3. For LangSplat and LEGaussians, we followed their operation to reconstruct the 512-dimensional CLIP feature from the low-dimensional language feature of each Gaussian. Note that our evaluations all follow a consistent setup: we use text to find matching 3D Gaussians, which are then rendered into multi-view images. Therefore, the metrics we report are inconsistent with the official metrics reported by comparison methods. **3) Dataset and Metrics**: We conducted experiments on the Lerf-ovs dataset re-annotated by LangSplat. The average IoU and accuracy are calculated between the images rendered from the 3D Gaussian points selected by the text query and the GT object masks.

**Results.** The quantitative results are shown in Tab. 1. The comparison methods exhibit poor 3D understanding capabilities, meaning they struggle to accurately identify the 3D Gaussian points relevant to the query text. We attribute this to the following reasons: **1)** Weak feature discrimination. Both LangSplat and LEGaussians compress high-dimensional CLIP features into low dimensions. Although these are then reconstructed using a decoder, the transformation is not lossless, reducing the distinctiveness between different semantic concepts and resulting in many similar features. **2)** The alpha-blending weighted accumulation rendering method cannot ensure a one-to-one correspondence between 2D image features and 3D point features, causing their 3D point-level performance to fall significantly short of their 2D pixel-level performance. Conversely, our method achieves superior performance by addressing the two issues faced by the comparison methods: **1)** We obtain distinctive features through semantic-agnostic feature learning (Sec. 3.1) and two-level codebook discretization (Sec. 3.1); **2)** We avoid the learning burden of high-dimensional CLIP features and ensure lossless features through training-free instance-level 2D-3D feature association (Sec. 3.3).

The visualization results are presented in Fig. 5. Given a query text, we can select the relevant Gaussian points and render them into multi-view images. However, the comparison methods make

Table 2: Performance of semantic segmentation on the Scannet dataset compared to LangSplat and LEGaussians based on text query.

| Methods | 19 classes | | 15 classes | | 10 classes | |
|---|---|---|---|---|---|---|
| | mIoU ↑ | mAcc. ↑ | mIoU ↑ | mAcc. ↑ | mIoU ↑ | mAcc. ↑ |
| LangSplat [33] | 3.78 | 9.11 | 5.35 | 13.20 | 8.40 | 22.06 |
| LEGaussians [37] | 3.84 | 10.87 | 9.01 | 22.22 | 12.82 | 28.62 |
| OpenGaussian | **24.73** | **41.54** | **30.13** | **48.25** | **38.29** | **55.19** |

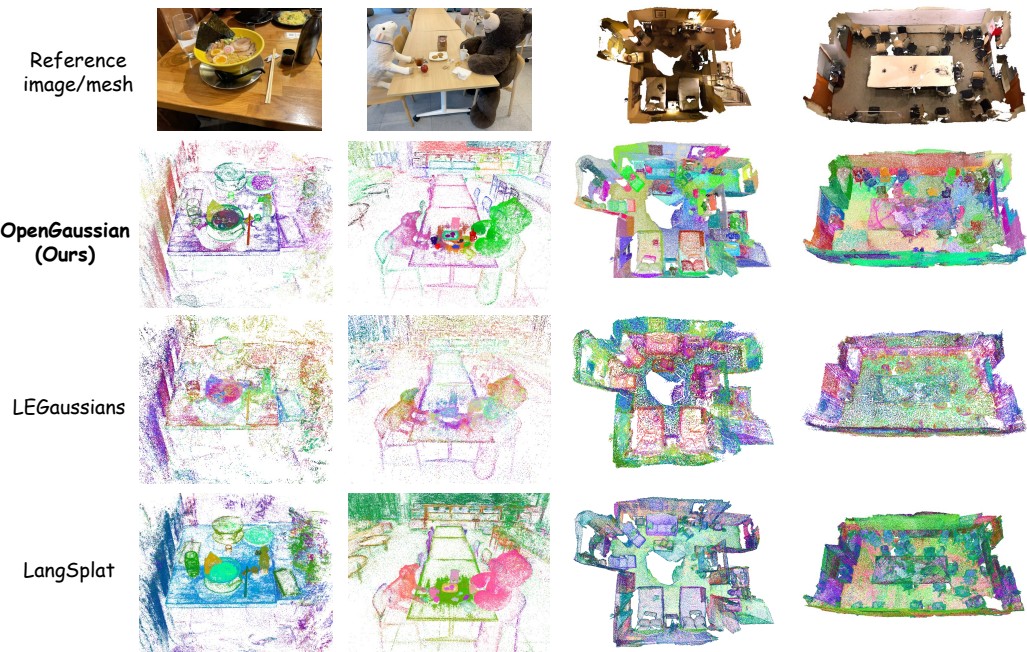

Figure 6: 3D feature visualization comparison. From left to right, the scenes are `ramen`, `teatime`, `scannet_0140_00`, and `scannet_0645_00`. Our proposed method, OpenGaussian, exhibits enhanced granularity and accuracy in its features.

it difficult to identify the accurate target due to the ambiguous 3D point features. In the left two columns of Fig. 6, we show the results of feature visualization on the LERF dataset. Our features exhibit better discrimination.

## 4.2 Open-Vocabulary Point Cloud Understanding

**Settings. 1) Task**: Given a set of open-vocabulary text queries, we calculate the cosine similarity between these text features and the Gaussian features. For each Gaussian, we select the text with the highest similarity as its category, constituting the open-vocabulary point cloud understanding task. **2) Baseline:** The comparison methods are consistent with those in the last section, *i.e.* LangSplat and LEGaussians. The high-dimensional feature reconstruction method for the Gaussian points is also the same. **3) Dataset and Metrics**: We conduct comparisons on the ScanNetv2 dataset [9], which provides posed RGB images from video scans, as well as reconstructed point clouds and GT 3D point-level semantic labels. Both our method and the comparison methods use the provided point clouds for initialization. During training, we *freeze the coordinates of the point clouds* and disable the densification process of 3DGS to ensure that the number and coordinates of the output point clouds match those of the input/GT point clouds. We randomly selected 10 scenes for evaluation, with training images extracted every 20 frames from the given video images. We use point cloud mIoU and mAcc as evaluation metrics.

**Results.** Tab 2 shows the performance when using 19, 15, and 10 categories from the ScanNetv2 dataset as text queries. The dataset provides a total of 19 semantic categories (excluding "other furniture"). Our method significantly outperforms the comparison methods. Notably, in our framework,

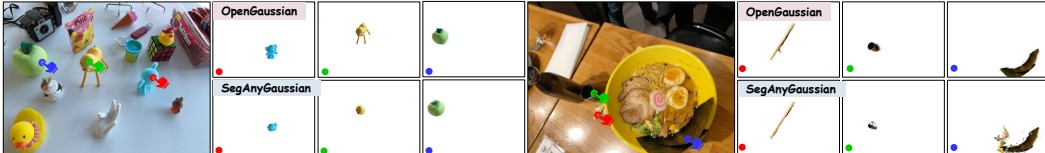

Figure 7: Subjective result of click-based 3D object selection. OpenGaussian demonstrates a more complete 3D object selection without the issues of incompleteness or redundancy.

CLIP features are utilized in a zero-shot manner without any training, whereas in the comparison methods, CLIP features are involved in learning Gaussian features. This highlights the low cost and high efficiency of our approach. The performance of the comparison methods on the ScanNet dataset is even lower than on the LeRF dataset (Tab 1). We attribute this mainly to the retention of the densification operation of 3DGS on the LeRF dataset, which involves millions of points per scene. In contrast, the point clouds provided by the ScanNet dataset are sparse, with approximately one hundred thousand points per scene. This sparsity means that in ScanNet scenes, a single point may need to represent the appearance of multiple pixels, exacerbating the issue of inconsistencies between 2D and 3D features and leading to poorer performance. The results of point cloud feature visualization on ScanNet are shown in the right two columns of Fig. 6. Our features exhibit better instance-level discrimination.

### 4.3 Click-based 3D Object Selection

SAGA [8] shares a similar motivation with our approach, learning distilled SAM features to support selecting 3D points associated with a 2D pixel clicked in the image. However, it lacks language-grounded ability. In contrast, ***our method does not require supervision from SAM features and can achieve click-based object selection*** using only the first two steps of our approach (Sec. 3.1, Sec. 3.2). Given an image from any viewpoint, click a 2D pixel and select the related 3D Gaussian points corresponding to that 2D pixel, then render them across multiple views. Unlike Sec. 4.1, where the query is text, the input for this experiment is the pixel coordinates of the clicked point. In Fig. 7, we compare our method with SAGA on the LERF dataset. The results show that our method can segment more complete 3D objects. It is worth noting that SAGA employs post-processing methods such as SAM mask, statistics, region growing, and ball query during inference.

### 4.4 Ablation Study

**(1) Ablation of Intra-mask Smooth Loss and Inter-mask Contrastive Loss.** To validate the effectiveness of the two losses proposed in Sec. 3.1, we conducted the ablation shown in Tab. 3. **i)** The inter-mask contrastive loss proves to be more crucial. Employing only this loss achieves respectable performance. Adding the intra-mask loss further enhances results, leading to a 3.05% improvement in mIoU and a 2.76% increase in mAcc. **ii)** The intra-mask smooth loss exhibits comparatively lower importance, which can be attributed to the inherent characteristics of 3DGS, where a single Gaussian point represents multiple pixels. Consequently, features of neighbouring pixels tend to be similar, indicating that 3DGS naturally induces a smoothing effect on adjacent pixels. This intrinsic smoothing mechanism partially mitigates the contribution of the intra-mask smooth loss.

**(2) The Necessity of the Two-Level Codebook. i)** In case #1 of Tab 5, a single-layer codebook with $k = 64$ was initially employed, resulting in a limited 28% mIoU. The primary limitation arose from the codebook's capacity, which was insufficient to represent all objects in the scene. Increasing $k$ to 320 in case #2 seemed like an intuitive solution, but it led to a significant decrease in performance. Visualizations highlighted that solely constraining instance features resulted in spatially distant points being grouped together within the same cluster. **ii)** To address this, we introduced a two-level codebook approach. At the coarse level, we utilized both instance features and coordinates to ensure spatial proximity of 3D points within clusters. Then, at the fine level, we further discretized instance features. Notably, case #5 demonstrated substantial performance improvements with the two-level codebook. In contrast, case #6, which employed a two-level design without incorporating coordinates at the coarse level, underscored the positive impact of including coordinates. **iii)** To illustrate the importance of considering both coordinates and the two-level codebook, we conducted experiments with the case #3 and case #4 configurations, in which only position information was considered without using the two-level codebook.

**(3) The Strategy for 2D-3D Feature Association.** In our instance-level 2D-3D feature association, we assessed the IoU between 3D instance renderings and SAM masks, as well as the distance between instance features and pseudo-features. Through an ablation study presented in Tab 4, we found that each strategy can independently achieve comparable performance. Case #1 demonstrates the effectiveness of our codebook discretization, enabling accurate 3D instance acquisition to support the IoU-based association strategy. Case #2 highlights the discriminative power and global consistency of our instance features, showing that feature-based matching alone can effectively associate objects. Case #3 confirms that considering both strategies simultaneously yields the best performance. All the ablations are evaluated on the semantic segmentation task of the 10 categories on ScanNet.

Table 3: Inter/Intra loss ablation.

| Case | Inter | Intra | mIoU $\uparrow$ | mAcc. $\uparrow$ |
|------|-------|-------|------|------|
| #1 | ✓ | | 35.24 | 52.43 |
| #2 | | ✓ | 25.89 | 42.76 |
| #3 | ✓ | ✓ | **38.29** | **55.19** |

Table 4: Association strategy ablation.

| Case | IoU | Feat. dis. | mIoU $\uparrow$ | mAcc. $\uparrow$ |
|------|-----|------------|------|------|
| #1 | ✓ | | 35.28 | 53.19 |
| #2 | | ✓ | 34.01 | 51.35 |
| #3 | ✓ | ✓ | **38.29** | **55.19** |

Table 5: Performance of semantic segmentation with various codebook configurations.

| Case | Coarse-level w/o xyz | Coarse-level w/ xyz | Fine-level | mIoU $\uparrow$ | mAcc. $\uparrow$ |
|------|------|------|------|------|------|
| #1 | ✓ ($k$=64) | | | 28.68 | 47.27 |
| #2 | ✓ ($k$=320) | | | 14.61 | 24.34 |
| #3 | | ✓ ($k$=64) | | 32.04 | 49.82 |
| #4 | | ✓ ($k$=320) | | 15.20 | 24.91 |
| #5 | ✓ ($k$=64) | | ✓ | 30.27 | 46.44 |
| #6 | | ✓ ($k$=64) | ✓ | **38.29** | **55.19** |

## 5  Conclusion

In this paper, we introduce a 3DGS-based open vocabulary understanding method for 3D point-level tasks. Existing methods excel at the pixel level but perform poorly at the 3D point level due to learning lossy features and 2D-3D feature inconsistencies. We addressed this by training instance features with 3D consistency using SAM masks and proposing a two-level codebook to discretize these features, achieving intra-object consistency and inter-object distinction. Finally, we enabled open vocabulary capability through lossless instance-level 2D–3D CLIP feature associations.

**Limitations**: (1) The geometric properties of the Gaussian (position, opacity, scale) are fixed. This may lead to inconsistencies between geometric representation and semantic content. We will consider joint optimization of instance features and geometric properties in future work. (2) The values of $k$ for the two-level codebooks are determined empirically. It is necessary to study scenario-specific adaptive values to optimize performance across diverse contexts. (3) We focus on 3D point-level understanding without considering the regression of object sizes to perform open-vocabulary 3D detection tasks [27, 6, 5, 46]. (4) Currently, we have not considered dynamic factors, which are common challenges in real-world applications. Integrating the proposed method with 4DGS [43, 13] would be meaningful.

## Acknowledgments

This work was supported by the National Natural Science Foundation of China (Grant No. 62372016) and the Guangdong Provincial Key Laboratory of Ultra High Definition Immersive Media Technology (Grant No. 2024B1212010006).

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

# A  Appendix

## A.1  Implementation Details

**(1) Training Strategy.** Consistent with LangSplat, we first pre-train the standard 3DGS for 30,000 steps. Subsequently, we freeze the Gaussian coordinates, scale, and opacity parameters, and train the instance features for 10,000 steps (ScanNet is 20,000 steps) and the two-layer codebook for 30,000 steps (ScanNet is 40,000 steps). The 2D-3D feature association step is training-free. The extraction methods for SAM masks and CLIP features also align with LangSplat. While LangSplat extracts three layers of SAM masks (small, middle, and large), our implementation uses only one layer (large).

**(2) Training Time.** We train each scene on a single 32G V100 GPU (with actual memory usage around 16 to 20G). For the LERF dataset, each scene takes around 200 images and trains for approximately 50 minutes. For the ScanNet dataset, each scene takes around 100-300 images (Sample every 20 frames from the original data and downsample by 2), and trains for approximately 15 minutes. The 2D-3D feature association step is a one-time computation, and no further computation is needed during inference. The association process takes around 1 minute.

**(3) ScanNet Dataset Evaluation.** We randomly selected 10 scenes from ScanNet for evaluation, specifically: `scene0000_00`, `scene0062_00`, `scene0070_00`, `scene0097_00`, `scene0140_00`, `scene0200_00`, `scene0347_00`, `scene0400_00`, `scene0590_00`, `scene0645_00`.
The 19 categories (defined by ScanNet) used for text query are respectively: `wall`, `floor`, `cabinet`, `bed`, `chair`, `sofa`, `table`, `door`, `window`, `bookshelf`, `picture`, `counter`, `desk`, `curtain`, `refrigerator`, `shower curtain`, `toilet`, `sink`, `bathtub`;
15 categories are without `picture`, `refrigerator`, `showercurtain`, `bathtub`;
10 categories are further without `cabinet`, `counter`, `desk`, `curtain`, `sink`.

**(4) Hyperparameters.** 1) The values of $k$ in the two-level codebook. In the ScanNet dataset, $k1 = 64, k2 = 5$ are used uniformly. In the LeRF dataset, for the `teatime` scene, $k1 = 32, k2 = 10$; for the other scenes, $k1 = 64, k2 = 10$. 2) The weights of the coordinates in the coarse-level codebook. In the ScanNet dataset, the weight is $1.0$. In the LeRF dataset, the weight for the `teatime` scene is $0.1$, while for the other scenes, the weight is $0.5$. 3) The weight of the intra-mask smoothing loss. In the `ramen` scene of LeRF, the weight is $0.01$; for the other scenes and ScanNet, the weight is $0.1$.

## A.2  More Results

### A.2.1  Scene editing

Fig. 8 demonstrates the scene editing capabilities of our method. Based on the original scene (Fig. 8 (a)) reconstructed with OpenGaussian, we can select objects for removal (Fig. 8 (b)), insertion (Fig. 8 (c)), or color modification (Fig. 8 (d)).

### A.2.2  Instance feature visualization

Fig. 10 shows the visualization results of rendering 3D point instance features into multi-view images. Fig. 12 presents the visualization results of 3D point features for more scenarios.

### A.2.3  Qualitative results of outdoor and real-world scenarios

Fig. 13 shows qualitative results of 3D instance features for 6 sequences in the Waymo outdoor dataset, demonstrating the capability to discretize large cases.

Fig. 14 presents the results of rendering 3D features into 2D feature maps, showcasing the ability to learn instance features with 3D consistency from coarse SAM supervision.

Fig. 15 illustrates the effectiveness of the two-stage codebook in outdoor scenes.

Furthermore, we conducted validations in the real-world scene. As depicted in Fig. 11, using an office scene captured by a mobile phone, the visualization of 3D points demonstrates that OpenGaussian achieved significant object discrimination in the real world.

### A.2.4 Text-to-3D Gaussian retrieval

Fig. 9 shows a demo of retrieving relevant Gaussians via text query, which is achieved by computing the cosine similarity between text features and the language features associated with 3D Gaussians (Sec. 3.3).

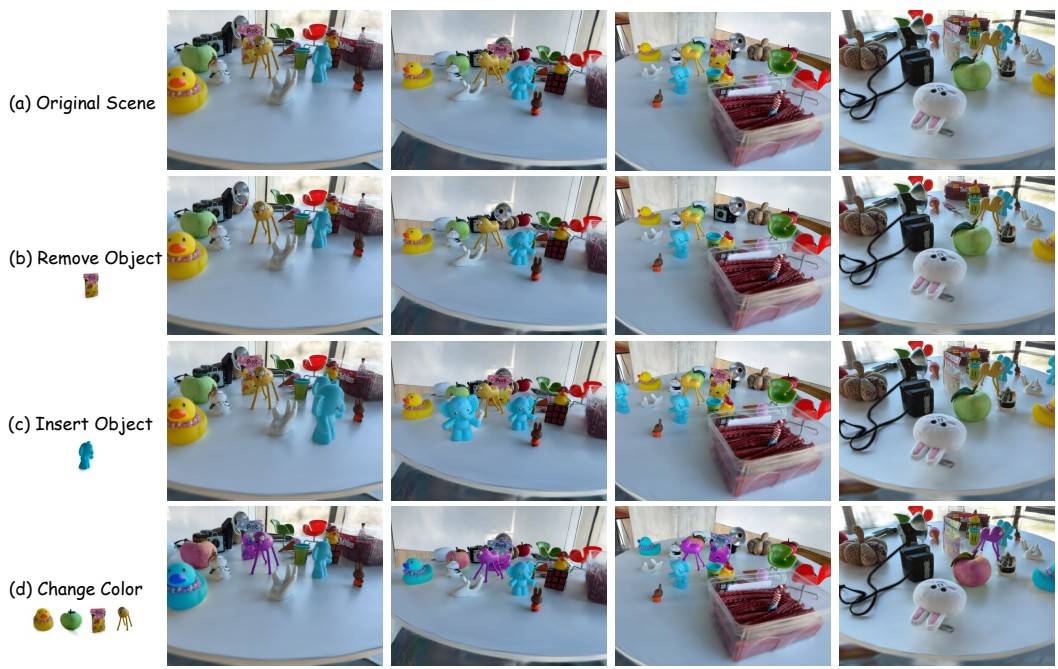

Figure 8: Examples of scene editing. (a) The original scene was reconstructed using OpenGaussian. (b) Selecting an object for removal. (c) Inserting a new object. (d) Changing the color of the selected object. Note that all edits are performed in 3D space, not on the image.

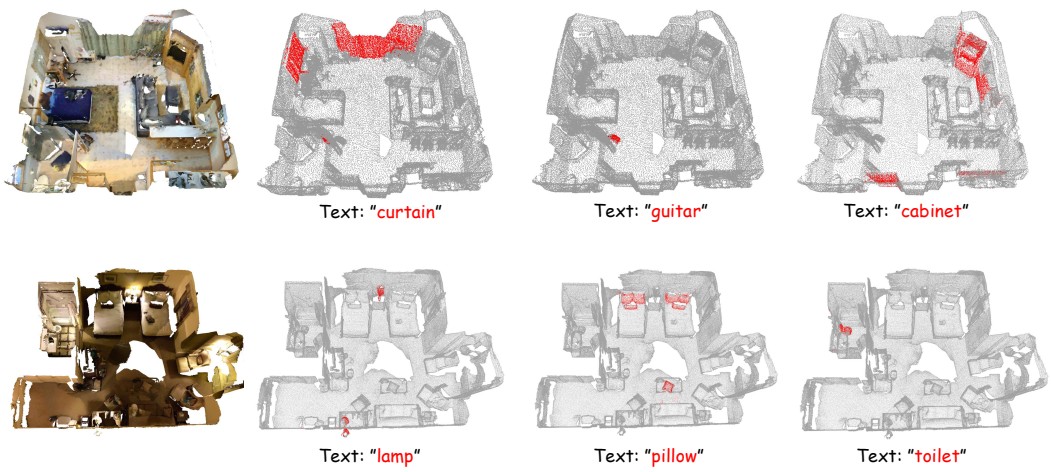

Figure 9: A demo of text-to-3D Gaussian retrieval on ScanNet. Top: `scene0000_00`; bottom: `scene0645_00`.

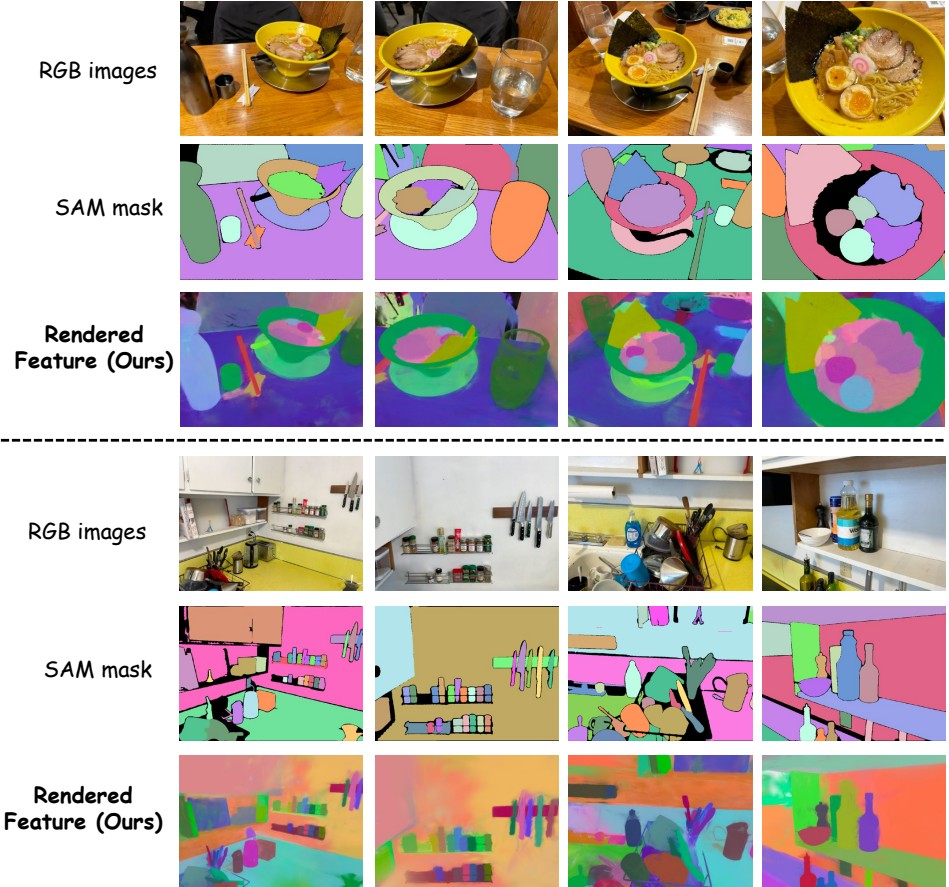

Figure 10: We rasterize the 3D point instance features into multi-view images, demonstrating cross-view consistency.

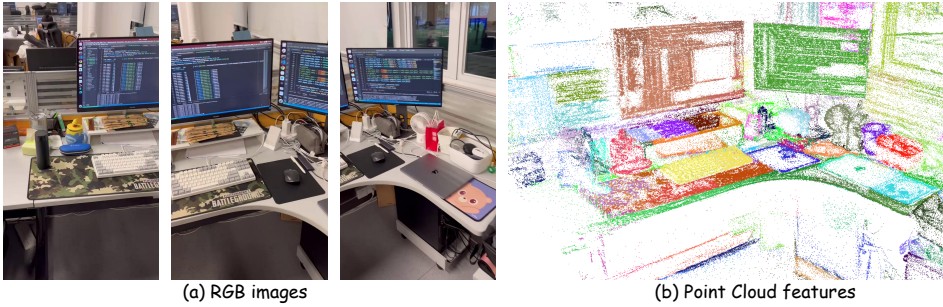

(a) RGB images                    (b) Point Cloud features

Figure 11: Visualization of 3D point features in the real-world scene captured by mobile phone.

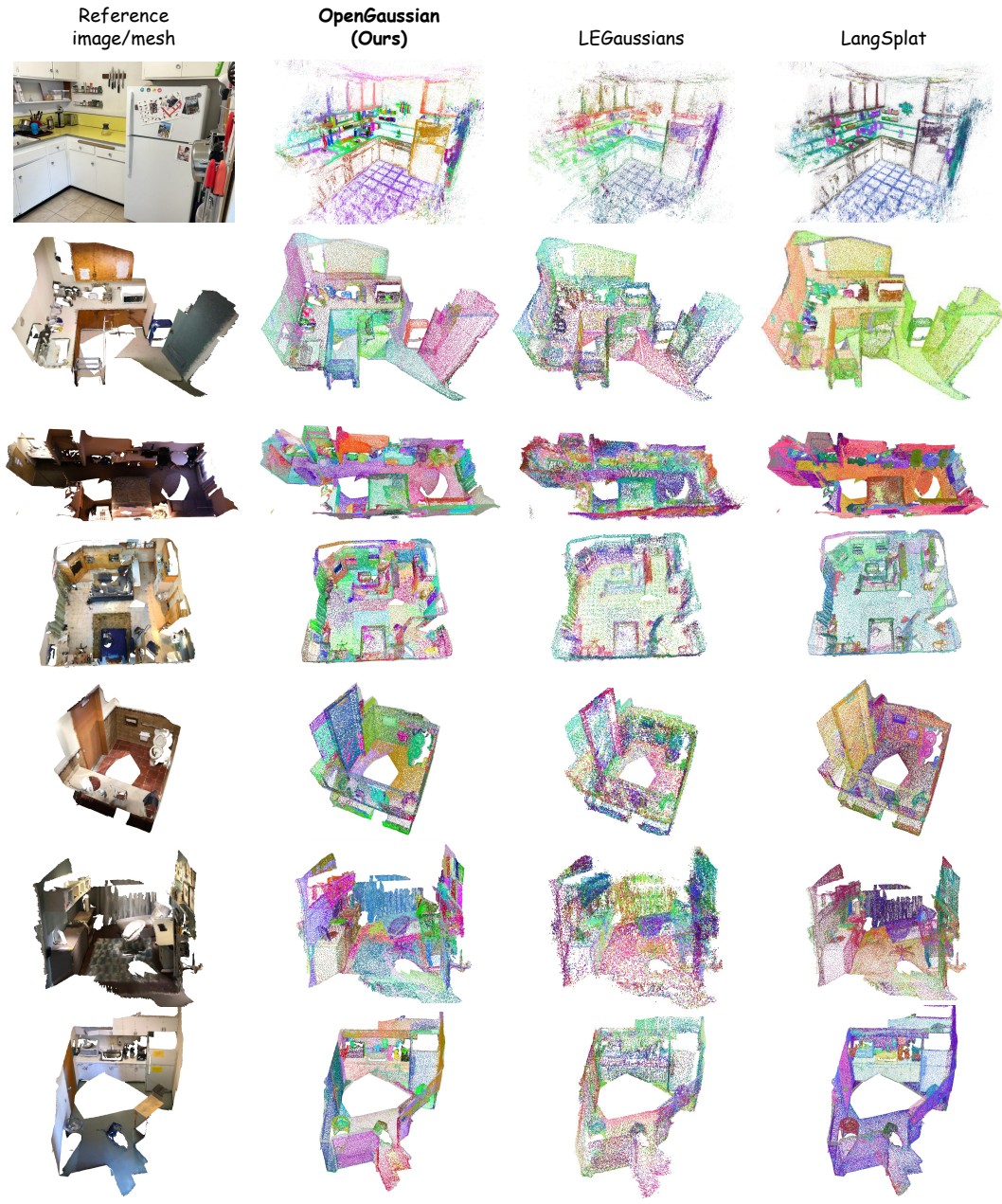

Figure 12: 3D Gaussian feature visualization on the LERF and ScanNet datasets.

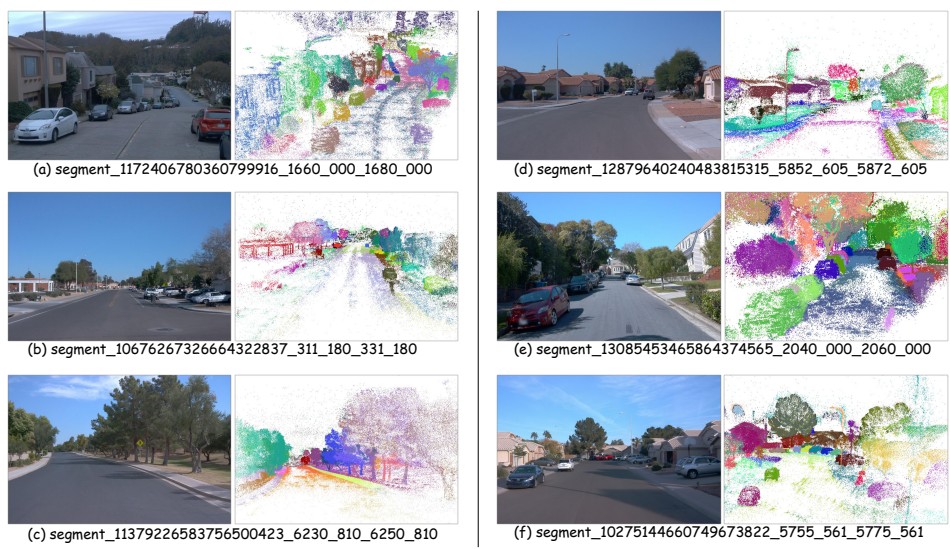

Figure 13: Feature visualization of 3D points on the large-scale outdoor dataset Waymo. (a)-(f) are 6 different scenes selected from the Waymo dataset. Left: RGB image; Right: 3D point features.

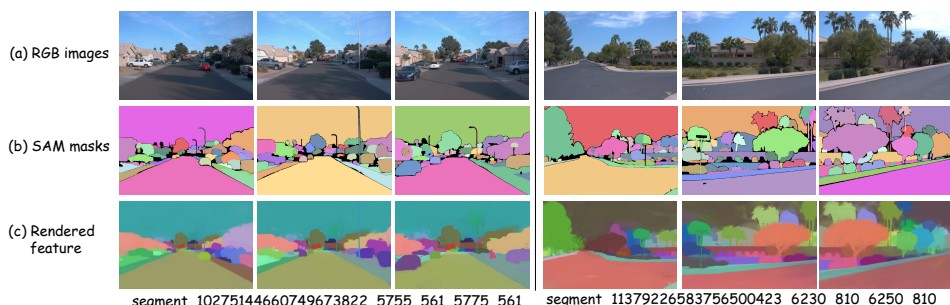

Figure 14: Results of rendering 3D point features onto images in the Waymo dataset. We trained 3D point features with multi-view consistency using SAM masks that without inter-frame associations.

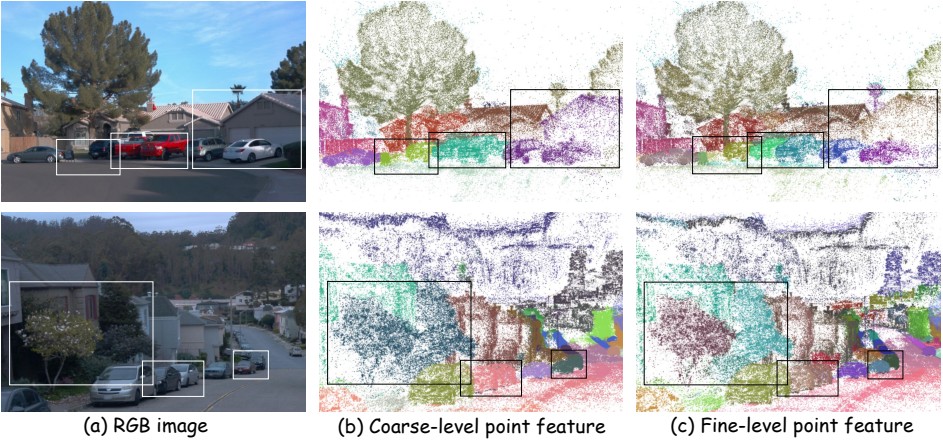

Figure 15: Validation of two-level codebook in outdoor scenes. Achieved better discretization with the fine-level codebook.

### A.3 Splatting 3D Gaussian Points

The 3D Gaussian point is formally defined as

$$G(\boldsymbol{x} \mid \boldsymbol{\mu}, \boldsymbol{\Sigma}) = e^{-\frac{1}{2}(\boldsymbol{x}-\boldsymbol{\mu})^{T}\boldsymbol{\Sigma}^{-1}(\boldsymbol{x}-\boldsymbol{\mu})} \tag{6}$$

In the given equation, $\boldsymbol{\mu} \in \mathbb{R}^3$ represents the spatial mean and $\boldsymbol{\Sigma} \in \mathbb{R}^{3\times3}$ denotes the covariance matrix. To ensure validity throughout the optimization process, the covariance matrix $\boldsymbol{\Sigma}$ is decomposed into a scaling matrix $\boldsymbol{S}$ and a rotation matrix $\boldsymbol{R}$ as follows:

$$\boldsymbol{\Sigma} = \boldsymbol{R}\boldsymbol{S}\boldsymbol{S}^{\top}\boldsymbol{R}^{\top} \tag{7}$$

During the rendering process, the 3D Gaussians are projected onto a 2D plane. With the intrinsic matrix $\boldsymbol{K}$ and extrinsic matrix $\boldsymbol{T}$, the 2D mean $\boldsymbol{\mu}'$ and covariance $\boldsymbol{\Sigma}'$ are defined as follows:

$$\boldsymbol{\mu}' = \boldsymbol{K}[\boldsymbol{\mu}, 1]^{\top}, \quad \boldsymbol{\Sigma}' = \boldsymbol{J}\boldsymbol{T}\boldsymbol{\Sigma}\boldsymbol{T}^{\top}\boldsymbol{J}^{\top} \tag{8}$$

Here, $\boldsymbol{J}$ represents the Jacobian of the affine approximation of the projective transformation. Each Gaussian is associated with an opacity value o and a view-dependent color $\boldsymbol{c}$, determined by a set of spherical harmonics coefficients. The pixel color $\boldsymbol{C}$ is computed by performing alpha-blending on the sorted 2D Gaussians, starting from the front and progressing toward the back.

$$\boldsymbol{C} = \sum_{i \in N} T_i G_i \left(\boldsymbol{u} \mid \boldsymbol{\mu}', \boldsymbol{\Sigma}'\right) \sigma_i \boldsymbol{c}_i \tag{9}$$

where $T_i = \prod_{j=1}^{i-1} \left(1 - G_i \left(\boldsymbol{u} \mid \boldsymbol{\mu}', \boldsymbol{\Sigma}'\right) \sigma_i\right)$.

