# OpenReview forum: "OpenGaussian: Towards Point-Level 3D Gaussian-based Open Vocabulary Understanding"
_NeurIPS.cc/2024/Conference — NeurIPS 2024 poster_

### Official Review · Reviewer_d9so · 2024-07-08

**Soundness:** 3
**Presentation:** 3
**Contribution:** 3
**Rating:** 6
**Confidence:** 3

**Summary:**

This paper presents OpenGaussian, a method utilizing 3D Gaussian Splatting for open vocabulary comprehension at the 3D point level. It addresses the limitations of current 3DGS methods that are confined to 2D pixel-level analysis and are inadequate for 3D point-level tasks. The method leverages SAM masks to maintain 3D consistency, implements a two-stage codebook for feature discretization, and proposes an instance-level 3D-2D feature association approach, demonstrating effectiveness in various tasks.

**Strengths:**

1. The motivation behind this paper is well-grounded, and the proposed method is technically sound.
2. The paper is well-written and easy to follow.
3. The method demonstrates good performance across various tasks, including open-vocabulary object selection, click-based 3D object selection, and 3D point cloud understanding.

**Weaknesses:**

1. The ablation studies are not sufficiently thorough. In Table 3, using xyz information results in significant performance improvement. It would be beneficial to provide results for the coarse-level alone, using xyz information, with k=64 and k=320. This would further illustrate the necessity of using the fine-level.
2. A sensitivity analysis for k1 and k2 could be provided to better understand their impact on the performance of the method.

**Questions:**

See weaknesses

**Limitations:**

Yes

---

> ### Author Rebuttal · Authors · 2024-08-06
>
> ## 1. More Detailed Ablation of Two-Stage Codebooks
> Thank you for your constructive comments. Following your suggestions, we have incorporated two additional experiments (refer to cases #3 and #4 in the table below) to evaluate the performance when considering xyz coordinate information at the coarse level alone. Our analysis yielded the following insights:
> + **Importance of xyz Information**: Comparing cases #1 and #3, we observe a significant improvement in mIoU (3.36%) and mAcc (2.55%) when incorporating xyz information at the coarse level. This highlights the crucial role of spatial context in our approach.
> + **Codebook Size**: Comparing cases #2 and #4, we find that simply increasing the number of codewords (k) does not necessarily lead to performance gains.
> + **Fine-Level Codebook Contribution**: Comparing cases #3 and #6, we see a substantial performance boost of 6.25% in mIoU and 5.37% in mAcc when incorporating the fine-level codebook. This demonstrates the significant contribution of the fine-level codebook in capturing detailed semantic information and enhancing the overall performance.
>
> The additional experiments reinforce our claim that incorporating xyz coordinate information at the coarse level is essential and that the fine-level codebook plays a critical role in boosting performance. We appreciate your suggestions, which have significantly strengthened the persuasiveness of our method.
>
> | Case |  Coarse-level    |        | Fine-level | mIoU  | mAcc. |
> |:----:|:--------------:|:-------------:|:---------:|:---:|:------:|
> |       | w/o xyz           | w/ xyz          |            |       |        |
> | #1   | ✓ (k=64)           |                |            | 28.68 | 47.27 |
> | #2   | ✓ (k=320)          |                |            | 14.61 | 24.34 |
> | #3   |                    | ✓ (k=64)       |            | 32.04 | 49.82 |
> | #4   |                    | ✓ (k=320)      |            | 15.20 | 24.91 |
> | #5   | ✓ (k=64)           |                | ✓          | 30.27 | 46.44 |
> | #6   |                    | ✓ (k=64)       | ✓          | 38.29 | 55.19 |
>
> ---
> ## 2. Sensitivity Analysis of k1 and k2
> Thank you for your question. In the ScanNet dataset, our default settings are k1=64 and k2=5. To address your concerns, we conducted a comprehensive analysis with k1 values of 48, 64, and 80, and k2 values of 3, 5, and 7, as illustrated in the table below. Our findings reveal that the optimal values for k1 and k2 should not be too high. Specifically, performance remains comparable when k1 is set to 64 or 48, and when k2 is set to 5 or 3. However, a performance degradation is observed when k1 and k2 are increased to 80 and 7.
>
> We sincerely appreciate your insightful suggestions, which led to these valuable observations. While our initially chosen values are empirically derived, these experiments highlight that setting fixed values may not always be optimal. In the future, we plan to investigate scene-adaptive k values, which could potentially enhance the performance across diverse scenarios.
>
> We commit to incorporating the aforementioned experiments and this limitations analysis in the revised version
>
> | Case          | k1  | k2 | mIoU | mAcc. |
> |:------------:|:--:|:--:|:-----:|:-----:|
> | #1            | 48   | 5   | 37.18  | 54.56 |
> | #2 (default)   | 64  | 5   | 38.29 | 55.19 |
> | #3            | 80   | 5   | 32.11  | 45.87 |
> | #4            | 64   | 7   | 34.89  | 49.33 |
> | #5            | 64   | 3   | 38.3   | 55.62 |

---

> > ### Comment · Reviewer_d9so · 2024-08-11
> >
> > Thank you for your response. My concern has been resolved, and I will maintain the original score.

---

> > > ### Author Response · Authors · 2024-08-11
> > >
> > > We greatly appreciate your response and valuable suggestions, which improved the quality and comprehensiveness of our paper.

---

### Official Review · Reviewer_sSBZ · 2024-07-11

**Soundness:** 3
**Presentation:** 2
**Contribution:** 3
**Rating:** 6
**Confidence:** 3

**Summary:**

In this paper, the authors propose three techniques to enhance the point-level 3D gaussian-based open vocabulary understanding:
1. Intra-mask smoothing loss to draw features within the same mask closer, and inter-mask contrastive loss to increase discriminativeness of the mean feature of each instance.
2. Two-level codebook for discretization. The proposed codebook discretizes instance features with the 3D coordinates to ensure identical Gaussian features from the same instance, in a coarse-to-fine manner.
3. Instance-level 2D-3D association technique to link CLIP features with 3D instance without loss backpropagation and depth information.

**Strengths:**

1. Visualizations are clear and strong to show the effectiveness of the proposed OpenGaussian and advantages over previous literature.
2. Quantitative results also demonstrate consistant and remarkable improvements over previous methods.

**Weaknesses:**

1. Paper writing needs to be improved. I have to admit that I'm not expert of this field, and this paper is not easy to understand since it requires abundant prior knowledge about the task and previous methods.
2. No limitation discussion is included in this paper.
3. Ablations on inter/intra-mask smoothing loss, since this is also a contribution of this paper.
4. Efficiency comparison between OpenGaussian and previous methods. The authors do includ training time in the supplemental material. However, since there is human-computer interaction in this task, it is critical to reveal the inference time or throughput of the method. Can it achieve real-time performance? Does it lags behind previous methods in terms of efficiency?
5. The three contributions of this paper seems separate. The authors are encouraged to resummarize them in a more integrated story.

**Questions:**

1. typo: m(m+1) -> m(m-1) in equation (2).

**Limitations:**

The paper doesn't discuss any limitations.

---

> ### Author Rebuttal · Authors · 2024-08-06
>
> ## 1. Ablation of Inter/Intra Mask Loss
> We truly appreciate your constructive feedback and apologize for any overlook regarding the ablation of the inter/intra mask loss. In response, we have conducted targeted ablation experiments, and the results are presented in the table below. Our analysis is summarized as follows:
> + The inter-mask contrastive loss proves to be more crucial. Employing only this loss achieves respectable performance. Adding the intra-mask loss further enhances results, leading to a 3.05% improvement in mIoU and a 2.76% increase in mAcc.
> + The intra-mask smooth loss exhibits comparatively lower importance: This can be attributed to the inherent characteristics of 3DGS, where a single Gaussian point represents multiple pixels. Consequently, features of neighboring pixels tend to be similar, indicating that 3DGS naturally induces a smoothing effect on adjacent pixels. This intrinsic smoothing mechanism partially mitigates the contribution of the intra-mask smooth loss.
>
> These additional experiments substantiate the contribution of our proposed inter/intra mask loss. We commit to incorporating these results and analysis in the revised version.
>
> | Case |   inter-mask loss  |   intra-mask loss  | mIoU  | mAcc. |
> |:----:|:----------------:|:-----------------:|:-----:|:-----:|
> | #1   | ✓               |                  | 35.24 | 52.43 |
> | #2   |                 | ✓                | 25.89 | 42.76 |
> | #3   | ✓               | ✓                | 38.29 | 55.19 |
>
> ---
> ## 2. Efficiency and Real-Time Performance Analysis
> We appreciate the reviewer’s question. Our experimental setup aligns with LangSplat and LEGaussian, focusing on semantic understanding of well-reconstructed 3D scenes, which is not a real-time task. However, we believe our method has potential for real-time applications for the following reasons.
> + **Incremental Input Support**: Unlike LangSplat[30] and LEGaussian[34], which necessitate acquiring all scene objects prior to training the autoencoder-decoder or features distillation, our method allows for the incremental input of new images. Each new frame can be processed immediately without prior preprocessing. This aligns well with the incremental needs of real-time tasks such as SLAM and robotics.
> + **Training Efficiency**: Our statistics on the ScanNet indoor dataset show that 3D point feature learning on a 640x480 image takes approximately 50ms per iteration for a scene with around 100,000 points, achieving a 20fps frame rate. On the Waymo outdoor dataset, processing a 960x640 image for a scene with approximately 700,000 points requires about 80ms per iteration, resulting in a 13fps frame rate. Furthermore, techniques like keyframes, sliding windows, and multithreading can be employed to further enhance processing speed in applications such as SLAM.
>
> The analysis suggests our method possesses the potential for real-time implementation. We also hope to inspire the community to utilize the proposed method across various downstream tasks by releasing our code publicly.
>
>
> ---
> ## 3. Discussion of Limitations
> We apologize for not analyzing the limitations of the proposed method. In the revised version, we will incorporate the following discussion:
> + The geometric properties of the Gaussian (position, opacity, scale) are fixed. This may lead to inconsistencies between geometric representation and semantic content. We will consider joint optimization of instance features and geometric properties in future work.
> + The values of k for the two-level codebooks are currently determined empirically. It is necessary to study scenario-specific adaptive values to optimize performance across diverse contexts.
> + Currently, we have not considered dynamic factors, which are common challenges in real-world applications. Integrating the proposed method with 4DGS would be meaningful.
>
> By acknowledging these limitations, we aim to provide a more balanced perspective on our method and suggest areas for future improvements.
>
>
> ---
> ## 4. Improvements in Paper Writing
> We appreciate the reviewer pointing out the shortcomings in our writing. Indeed, our contribution 1 (inter/intra-mask loss for instance feature learning) and contribution 2 (two-level codebooks for discrete feature learning) are closely linked, with contribution 1 essentially serving as the initialization for contribution 2. As for contribution 3, we do realize that this part is somewhat independent, leading to difficulty in understanding. We apologize for any confusion caused. In the revised version, we will enhance the readability of the paper in the following ways:
> + Reorganizing the introduction to better connect the three methods we proposed.
> + Adding contextual transitions between sections.
>
> ---
> ## 5. Other Question
> Thank you for pointing out the typo in Eq(2). We will correct this in the revised version.

---

> > ### Comment · Reviewer_sSBZ · 2024-08-12
> > **Response to the Rebuttal**
> >
> > Thanks for the authors' comprehensive rebuttal to my questions and sorry for the late reply. Their clarification on missing ablation studies and real-time performance analysis meet my satisfaction. Therefore, I will raise my rating to 6.
> >
> > However, I have to admit that I'm not an expert in this field, so please consider more about the review opinions of reviewers with higher confidence.

---

> > > ### Author Response · Authors · 2024-08-12
> > >
> > > We are very grateful for your response and the positive evaluation of our work. Your suggestions have significantly improved our manuscript, and we will also actively consider the opinions of other reviewers.

---

> ### Author Response · Authors · 2024-08-12
>
> Dear Reviewer sSBZ,
>
> Thank you once again for your insightful review, which has greatly enhanced the quality and clarity of our paper. We sincerely hope that our rebuttal has effectively addressed your questions and concerns. Should you require any additional clarifications or further information, please do not hesitate to reach out. We greatly value your insightful suggestions.
>
>
> Thank you very much for your time and consideration.
>
> Best regards,
>
> Authors of Submission 1591

---

### Official Review · Reviewer_ANGY · 2024-07-13

**Soundness:** 3
**Presentation:** 3
**Contribution:** 3
**Rating:** 6
**Confidence:** 5

**Summary:**

This paper introduces "OpenGaussian," a novel method for 3D point-level open vocabulary understanding using 3D Gaussian Splatting (3DGS). The authors address the limitations of existing 3DGS-based methods that primarily focus on 2D pixel-level parsing. OpenGaussian aims to enhance 3D point-level understanding by training instance features with 3D consistency and proposing a two-stage codebook for feature discretization. The method also introduces an instance-level 3D-2D feature association to link 3D points to 2D masks and CLIP features. Extensive experiments demonstrate the effectiveness of OpenGaussian in various 3D tasks, and the source code will be released.

**Strengths:**

- **Novelty**: The paper introduces a unique approach to 3D point-level open vocabulary understanding, which is a significant advancement over existing methods that focus on 2D pixel-level parsing.
 &nbsp;
- **Technical Contributions**: The proposal of a two-stage codebook for feature discretization and the introduction of a 3D-2D feature association method are innovative and well-executed.
 &nbsp;
- **Experiments**: The extensive experiments, including open vocabulary-based 3D object selection and 3D point cloud understanding, validate the effectiveness of the proposed method.
 &nbsp;
- **Clarity**: The paper is well-written and clearly explains the methodology, making it easy to follow the proposed approach and its benefits.

**Weaknesses:**

- **Limitations Discussion**: The paper does not discuss the limitations of the proposed method in detail, which could provide a more balanced view of its applicability and potential drawbacks.
 &nbsp;
- **Comparative Analysis**: While the paper compares OpenGaussian with LangSplat and LEGaussians, additional comparisons with other state-of-the-art methods in OV 3D understanding could strengthen the evaluation, like Open-vocabulary 3D object detection[1, 2]
 &nbsp;
- **Complexity**: The implementation details, especially the two-stage codebook and feature association, may be complex and could benefit from further simplification or more detailed explanations for reproducibility.
 &nbsp;
- **Generalization**: The experiments are conducted on specific datasets, and it's unclear how well the method generalizes to other types of 3D scenes or datasets.

[1] Yuheng Lu, Chenfeng Xu, Xiaobao Wei, Xiaodong Xie, Masayoshi Tomizuka, Kurt Keutzer, and Shanghang Zhang. Open-vocabulary point-cloud object detection without 3d annotation. In CVPR, 2023. 1, 3
[2] Yang Cao, Zeng Yihan, Hang Xu, and Dan Xu. Coda: Collaborative novel box discovery and cross-modal alignment for open-vocabulary 3d object detection. In NeurIPS, 2023

**Questions:**

- **Scalability**: How does the method perform on larger and more complex 3D scenes? Are there any scalability issues?
 &nbsp;
- **Real-time Performance**: Can the method be applied in real-time applications, especially in robotics and embodied intelligence scenarios?
 &nbsp;
- **Ablation Studies**: Can you provide more detailed ablation studies to isolate the contributions of each component of the proposed method?
 &nbsp;
- **Generalization**: Have you tested the method on different types of 3D datasets to evaluate its generalization capabilities?

**Limitations:**

The authors did not provide an analysis of the limitations. For suggestions on improvement, please refer to the weaknesses section.

---

> ### Author Rebuttal · Authors · 2024-08-06
>
> ## 1. Scalability and Generalization
> We appreciate the reviewer’s insightful question, which encouraged us to explore the scalability and generalizability of OpenGaussian in other 3D datasets and scenarios.
> + We selected **6 scenes from the Waymo outdoor dataset** captured by vehicle-mounted cameras to demonstrate the effectiveness of the proposed method in large-scale complex scenarios.
> + We captured images of a **real-world office scene** using a mobile phone to demonstrate the method’s generalization.
>
> Please refer to the **attached PDF** in the General Rebuttal section on OpenReview for detailed results. We kindly request the reviewer to check the file:
> + Fig.1, Fig.2: Visualization of 3D point features in 6 scenes from the Waymo dataset and the real-world office scene;
> + Fig.3: Results of rendering 3D point features into 2D feature maps;
> + Fig.4: Comparison of 3D point features after coarse-level and fine-level codebook discretization.
>
> We hope the added results can address your concerns about scalability and generalizability.
>
> ---
> ## 2. More Detailed Ablation
> Thank you for your question. Due to the interdependent nature of the three proposed methods, we cannot fully isolate them for ablation; the mask-based instance feature learning is essential for the two-level codebook discretization, and the discretized features are necessary for the 2D-3D association. However, we conducted a detailed ablation on their **subcomponents**, as shown in the table below. Cases #2, #3, and #4 were added during the rebuttal period.
> + Cases #2(w/o intra-mask) & #3(w/o inter-mask): Ablation of inter-mask and intra-mask losses, highlighting the importance of inter-mask loss.
> + Cases #1(two-level) & #4(coarse-level): Ablation of two-level codebooks, emphasizing the significance of fine-grained codebooks.
> + Cases #5(w/o feat. dis.) & #6(w/o IoU): Testing the 2D-3D feature association strategy, demonstrating the superiority of combined strategies.
> | Case|Inter/Intra-mask loss || Two-level codebook || 2D-3D association || mIoU  | mAcc. |
> |:-:|:-:|:-:|:-:|:-:|:-:|:-:|:-:|:-:|
> ||Inter|Intra|Coarse|Fine|IoU|Feat.Dis.|||
> |#1|✓|✓|✓|✓|✓|✓|38.29|55.19|
> |#2|✓||✓|✓|✓|✓|35.24|52.43|
> |#3||✓|✓|✓|✓|✓|25.89|42.76|
> |#4|✓|✓|✓||✓|✓|32.04|49.82|
> |#5|✓|✓|✓|✓|✓||35.28|53.19|
> |#6|✓|✓|✓|✓||✓|34.01|51.35|
>
> ---
> ## 3. Efficiency and Real-Time Performance Analysis
> We appreciate the reviewer’s question. Our experimental setup aligns with LangSplat[30] and LEGaussians[34], focusing on semantic understanding of well-reconstructed 3D scenes, which is not a real-time task. However, we believe our method has the potential for real-time applications for the following reasons.
> + **Incremental Input Support**: Unlike LangSplat and LEGaussian, which necessitate acquiring all scene objects prior to training the autoencoder-decoder or features distillation, our method allows for the incremental input of new images. Each new frame can be processed immediately without preprocessing. This aligns well with the incremental needs of real-time tasks such as SLAM and robotics tasks.
> + **Training Efficiency**: Our statistics on the ScanNet indoor dataset show that 3D point feature learning on a 640x480 image takes approximately 50ms per iteration for a scene with around 100,000 points, achieving a 20fps frame rate. On the Waymo outdoor dataset, processing a 960x640 image for a scene with approximately 700,000 points requires about 80ms per iteration, resulting in a 13fps frame rate. Furthermore, techniques like keyframes, sliding windows, and multithreading can be employed to further enhance processing speed in applications such as SLAM.
>
> The analysis suggests our method possesses the potential for real-time implementation. We also hope to inspire the community to utilize the proposed method across various downstream tasks by releasing our code publicly.
>
> ---
> ## 4. Comparison with 3D Open Vocabulary Detection Methods
> Thank you for raising this interesting point. In fact, these two types of methods differ significantly in their setups.
>
> The two papers mentioned by the reviewer require training a 3D Encoder-Decoder Backbone and a task-specific head, meaning they are data-driven and need to be trained on large-scale point cloud datasets. In contrast, our method follows the 3DGS setup, involving no networks (MLP, convolution, or Transformer) and only training point features. Besides, our method is scene-specific and not data-driven, thus avoiding domain gaps. The differences in model, training approach, and data make fair comparison challenging.
>
> Following your suggestion, we will include a discussion on these data-driven 3D open vocabulary understanding methods in the revised version.
>
>
> ---
> ## 5. Discussion of Limitations
> We apologize for not analyzing the limitations of the proposed method. In the revised version, we will incorporate the following discussions:
> + The geometric properties of the Gaussian (position, opacity, scale) are fixed. This may lead to inconsistencies between geometric representation and semantic content. We will consider joint optimization of instance features and geometric properties in future work.
> + The values of k for the two-level codebooks are currently determined empirically. It is necessary to study scenario-specific adaptive values to optimize performance across diverse contexts.
> + Currently, we have not considered dynamic factors, which are common challenges in real-world applications. Integrating the proposed method with 4DGS would be meaningful.
>
> By acknowledging these limitations, we aim to provide a more balanced perspective on our method and suggest areas for future improvements.
>
>
> ---
> ## 6. Implementation Complexity
> We apologize for any confusion caused. We commit to releasing the code for reproducibility and to contribute to the community. To enhance the clarity of the paper, we will provide more detailed implementation details in the revised version and simplify any non-essential components that may cause confusion.

---

> ### Author Response · Authors · 2024-08-12
>
> Dear Reviewer ANGY,
>
> Thank you once again for your insightful review, which has greatly enhanced the quality and clarity of our paper. We sincerely hope that our rebuttal has effectively addressed your questions and concerns. Should you require any additional clarifications or further information, please do not hesitate to reach out. We greatly value your insightful suggestions.
>
> Thank you very much for your time and consideration.
>
> Best regards,
>
> Authors of Submission 1591

---

> > ### Comment · Reviewer_ANGY · 2024-08-13
> > **Further reply**
> >
> > Dear authors, I have carefully read your paper. Most of my questions are addressed well. I have one more question. Is the feature learning process jointly conducted with the 3DGS learning? Or you learned the 3DGS firstly, then learn the instance/semantic feature further?

---

> > > ### Author Response · Authors · 2024-08-13
> > >
> > > Thanks for your feedback and positive evaluation of our work. We are glad that our rebuttal addressed your most concerns.
> > >
> > > For this question, we mentioned this training detail in "Implementation Details" of the supplementary material: For a fair comparison, we adhered to the training strategy consistent with LangSplat. We first trained for 30,000 steps using 3DGS, then froze the geometric properties of the Gaussians, and continued training the instance features. The advantage of this strategy is that it allows us to continue training from any model pre-trained on 3DGS (or its variants) without needing to retrain the geometric properties from scratch. However, as we noted in the first point of our limitations analysis (Rebuttal-Q5), this strategy may lead to inconsistencies between geometry and semantics. We appreciate your insightful observation, and we will further explore how to conduct more efficient joint training in the future.
> > >
> > > Thank you again for your thorough review. We would appreciate if you could consider re-evaluating our work in light of these clarifications.

---

> > > > ### Comment · Reviewer_ANGY · 2024-08-13
> > > > **Final score**
> > > >
> > > > Thanks to the authors for the quick reply and the efforts in the rebuttal, and sorry for missing the training detail. Please update the contents we discuss in the final version, i.e., the comparison with 3D Open Vocabulary Detection methods[1, 2] and the limitations. Totally speaking, most of my concerns are addressed well. So I raise my score to 6. Good luck :)

---

> > > > > ### Author Response · Authors · 2024-08-13
> > > > >
> > > > > We are very grateful for your recognition of our work and for the time and effort you invested in reviewing it. Your valuable suggestions have significantly enhanced the quality and scalability of our work. We will update these discussions in the revised version. Thank you again for your feedback.
> > > > >
> > > > > Best wishes!

---

### Author Rebuttal · Authors · 2024-08-07

Dear Reviewers and AC,

We would like to thank the three anonymous reviewers and the AC for their time and effort in reviewing our paper and providing constructive feedback. We are very grateful for the positive comments from the reviewers, such as “significant advancement over existing methods that focus on 2D pixel-level parsing” (Reviewer ANGY), “method are innovative and well-executed” (Reviewer ANGY), “extensive experiments” (Reviewer ANGY), “well-written and clearly explains” (Reviewer ANGY), “visualizations are clear and strong” (Reviewer sSBZ), “consistant and remarkable improvements” (Reviewer sSBZ), “motivation behind this paper is well-grounded” (Reviewer d9so), “well-written and easy to follow” (Reviewer d9so), and “good performance across various tasks” (Reviewer d9so).

For the insightful questions, constructive suggestions, and additional experiments requested by the reviewers, we have provided detailed responses to each reviewer. **Please refer to our individual responses to each reviewer for more details**. Below is a brief overview of our rebuttal content.

**Attached PDF**: Experiments on large-scale outdoor dataset Waymo and images from a real-world scene captured by the mobile phone to verify the scalability and generalization of the proposed method.

For **Reviewer ANGY**:
+ (1) **Scalability and Generalization**: We demonstrate the scalability and generalization of our method by adding experiments on the large-scale outdoor dataset Waymo and a real-world scene captured by the mobile phone. See the attached PDF for details.
+ (2) **More Detailed Ablation**: We conducted a comprehensive ablation study on each sub-component of our method to illustrate the role of each module.
+ (3) **Efficiency and Real-Time Performance Analysis**: We analyzed the real-time performance of the proposed method.
+ (4) **Comparison with 3D Open Vocabulary Detection Methods**: We discussed our method in comparison to data-driven 3D open vocabulary methods.
+ (5) **Discussion of Limitations**: We discussed the limitations of the proposed method.
+ (6) **Implementation Complexity**: We explained the complexity of the implementation and outlined ways to improve it.

For **Reviewer sSBZ**:
+ (1) **Ablation of Inter/Intra Mask Loss**: We added ablation experiments and analysis for both losses.
+ (2) **Efficiency and Real-Time Performance Analysis**: We analyzed the real-time performance of the proposed method.
+ (3) **Discussion of Limitations**: We discussed the limitations of the proposed method.
+ (4) **Improvements in Paper Writing**: We analyzed and improved the writing deficiencies.

For **Reviewer d9so**:
+ (1) **More Detailed Ablation of Two-Stage Codebooks**: We provided relevant ablation experiments and analysis to enhance the comprehensiveness and fairness of the ablation study.
+ (2) **Sensitivity Analysis of k1 and k2**: We conducted experiments and analysis to investigate the sensitivity of the codebook parameter.

We would like to thank the reviewers again for their valuable feedback, which has significantly improved the quality and comprehensiveness of our method. We hope our responses have addressed the reviewers’ concerns. If the reviewers have any further questions, we are more than happy to provide clarification.

Best regards.

---

### Decision · Program_Chairs · 2024-09-25

**Decision:**

Accept (poster)

**Comment:**

This paper introduces "OpenGaussian," a novel method for 3D point-level open vocabulary understanding using 3D Gaussian Splatting (3DGS). OpenGaussian aims to enhance 3D point-level understanding by training instance features with 3D consistency and proposing a two-stage codebook for feature discretization. The method also introduces an instance-level 3D-2D feature association to link 3D points to 2D masks and CLIP features. Experimental results are impressive. The final ratings are three weak acceptance. Given the consensus, the AC recommend acceptance as well.